# How Does Corporate Charitable Giving Affect Enterprise Innovation? A Literature Review and Research Directions

Lei Xu [1] , Xiaoning Guo [1], Yan Liu [1,*], Xiaochen Sun [2] and Jie Ji [3]

1 Economics and Management College, Civil Aviation University of China, Tianjin 300300, China
2 College of Management and Economics, Tianjin University, Tianjin 300072, China
3 School of Management, Tianjin University of China, Tianjin 300384, China
* Correspondence: lyailte@163.com; Tel.: +86-133-2406-7926

**Abstract:** During the past decades, academics and practitioners have basically reached a consensus on the relationship between corporate charitable giving and enterprise innovation. However, so far, few research studies have explored the essential reasons and the influencing mechanisms behind the relationship, through sufficient theoretical or empirical study. To clarify this relationship, this paper collects and reviews 196 related articles which include the topics of corporate philanthropy and corporate innovation over the period of 1966–2022, and analyzes their content in a systematic and comprehensive manner. Based on the literature analysis, it clarifies the current state and the differentiation trends of research topics in this field, then discusses the industry operations and theoretical development of corporate charitable giving. Through literature reviewing and practical analysis, this paper summarizes the mechanism and path of the influence of charitable giving on enterprise innovation from four perspectives: technical network, political reputation, media attention and resource adjustment, and proposes the external factors at macro- and meso-levels. By constructing a multi-level theoretical framework, this paper clarifies the preconditions, key factors and mediating mechanisms of charitable donation affecting enterprise innovation. This paper contributes a novel theoretical perspective for further theoretical development and for effectively managing corporate charitable giving and enterprise innovations. The paper concludes by offering several suggestions for future research on corporate charitable giving.

**Keywords:** corporate social responsibility; corporate charitable giving; enterprise innovation; theoretical framework

## 1. Introduction

Innovation is the cornerstone of business development for them to further their missions and carry out their operations. As the market becomes increasingly competitive, innovation practices will be considered as a prerequisite for companies in order to enhance their commercial value. Meanwhile, it is the responsibility and obligation of companies in the social environment to conduct charitable-giving practices. Corporate charitable giving can not only bring strategic resources and the conduit of information needed for enterprise innovation, but can also help companies enhance their moral capital, enhance their brand image, and enhance their political legitimacy, etc. Corporate charitable giving refers to the unconditional provision of funds or materials by companies to the government or relevant institutions in a voluntary and non-reciprocal way, solving social problems such as poverty, education, natural disasters, public health, etc. (FASB, 1993) [1]. As expectation and pressure from stakeholders in the marketplace continue to increase, charitable giving has become one of the indispensable driving forces of corporate growth (Gautier and Pache, 2015) [2].

During the past few decades, a substantial amount of research has been conducted concerning the impact of corporate charitable giving on enterprise innovation. Academics

have carried out theoretical and empirical studies on this issue from the aspects of influencing factors, mechanisms and effects etc. (Zhengang Zhang et al., 2016; Bereskin et al., 2016; Bereskin and Hsu, 2016) [3–5]. Some studies suggest a positive linear relationship between the impact of charitable giving on firm performance (Wang and Qian, 2011) [6], while others believe that there is a negative linear relationship (Brammer et al., 2006) [7]. Still some others explore a U-shaped relationship (Brammer and Millington, 2005) [8]. However, neither of them sorts out the specific relationship between corporate charitable giving and enterprise innovation, nor carry out the influence mechanism about this relationship.

In order to clarify the influence of corporate charitable giving on innovation development and to have a better understanding of the influence mechanism, this paper collected the Web of Science database with the term "corporate charitable giving" and obtained 196 related papers from 1966 to 2022. As a result of compiling and summarizing these 196 papers, we find that both theoretical and empirical literature on corporate philanthropy has developed significantly since the early 1990s. Multiple research studies on corporate charitable giving have explored the necessity of corporate charitable giving (Young and Burlingame, 1996) [9], the antecedents, mediators, and consequences of corporate charitable giving, as well as various interdisciplinary theoretical perspectives. While previous corporate charitable giving reviews have provided a helpful theoretical foundation to describe the development of this field, their contribution still had some limitations on the impact of corporate philanthropy on enterprise innovation and its mechanism in the academic field. Based on this, the purpose of this paper is to deconstruct corporate-charitable-giving behavior from the perspective of industry practice and theoretical development. Meanwhile, we also examine the mechanisms and paths of charitable giving influencing innovation from different perspectives. Through a multi-level theoretical framework, we hope to contribute to further interdisciplinary research on corporate philanthropy, and to guide both researchers and practitioners.

## 2. Methodology

In order to increase the validity of the findings for the paper, this paper is based on the process of a systematic literature review, which followed the systematic research process conducted by Bekkers and Wiepking (2010) [10]. Firstly, the researchers examined papers published in various fields related to corporate charitable giving. Major commercial databases were searched, such as ScienceDirect, ProQuest, Springer Link, and Emerald. We used the following keywords: donations, philanthropy, charitable giving and enterprise innovation; the search terms and other criteria were the same for all databases. Such a process resulted in the selection of 233 corporate philanthropy documents, which consist of 221 journal articles, 7 dissertations, and 5 books and book chapters. Secondly, we searched the authors' own databases and the references cited in the papers we found, to serve as a foundation in our search. Thirdly, we selected the search results and checked the changes of common terms and related words in different parts of published papers, including keywords, titles, research content and conclusions, and deleted articles that were not related to the research topic of this paper to improve the rigor and reliability of the search results. To make sure, we adopted a conservative or fine-grained approach toward the inclusion of CSR studies (Cha and Rajadhyaksha, 2021) [11]. We chose those findings that were most relevant in furthering our knowledge of charitable giving and enterprise innovation. Studies that covered variables related to corporate charitable giving were included; studies that were not related to corporate charitable giving or were lacking in theoretical or empirical contributions were excluded. Finally, we obtained 196 papers and documents published in 99 academic journals from 1966 to 2022.

### 2.1. Data Characteristics

In recent years, scholars have proposed, researched, and explored corporate philanthropic giving from different fields and achieved remarkable results. Based on the research questions of each paper, we refine the research scope of the literature and divide all the

literature into seven research topics according to different types of research, which are Corporate Charitable Giving and Performance, Corporate Charitable Giving and Innovation, Corporate Charitable Giving and Corporate Social Responsibility, Factors Influencing Corporate Charitable Giving, the Impact of Corporate Charitable Giving on Business, some Literature Reviews related to Corporate Charitable Giving, and others. In line with the above themes, we classified the main published literature from the past decades. The number and proportion of different study components are shown in Table 1. Although the research topics of these documents are divided into seven areas, these topics are often interrelated, and there may be multiple research topics in these documents at the same time. Therefore, the division of this paper is based on the research focus of each paper, so as to obtain more accurate data.

From the 196 papers and documents collected, we can find that both the two fields of the Impact of Corporate Charitable Giving on Business and Corporate Charitable Giving and Corporate Social Responsibility papers are important, accounting for a relatively large percentage, that is, 26.53% and 23.47%, respectively. Similarly, 30 papers are related to Factors Influencing Corporate Charitable Giving, accounting for 15.31% of the study. In addition, there are 19 papers related to Corporate Charitable Giving and Performance, 7 papers related to Corporate Charitable Giving and Innovation, 4 papers with some Literature Reviews related to Corporate Charitable Giving. In addition to all six fields, there are 38 papers in Others, accounting for 19.39%, including research on the impact of consumers on corporate charitable giving, case studies on corporate charitable giving, and research on market structure and corporate charitable giving.

**Table 1.** Distribution of papers in different fields [1–192].

| Application Fields | Number of Paper | Percentage |
|---|---|---|
| Corporate Charitable Giving and Performance | 19 | 10.20% |
| Corporate Charitable Giving and Innovation | 7 | 3.06% |
| Corporate Charitable Giving and Corporate Social Responsibility | 46 | 23.47% |
| Factors Influencing Corporate Charitable Giving | 30 | 15.31% |
| The Impact of Corporate Charitable Giving on Business | 52 | 26.53% |
| Some Literature Reviews related to Corporate Charitable Giving | 4 | 2.04% |
| Others | 38 | 19.39% |
| **Total** | **196** | **100%** |

## 2.2. Data Generalization

By further summarizing and analyzing the literature, we have summarized Figures 1 and 2. As shown in Figure 1, research on corporate charitable giving emerged in 1966, saw substantial growth after 1996, and has been a hot research issue for relevant scholars to this day. Figure 1 also shows the trend of the number of papers related to corporate charitable giving. Figure 2 further shows that the research on corporate charitable donation has gradually become mainstream within research. For example, a total of 25.5% of the corporate charity giving articles collected were published in *Journal of Business Ethics*, 11.2% in *Sustainability*, and so on, followed by *Journal of Sport Management*, *Organization Science*, *Journal of Management Studies*, *Nonprofit and Voluntary Sector Quarterly*, etc.

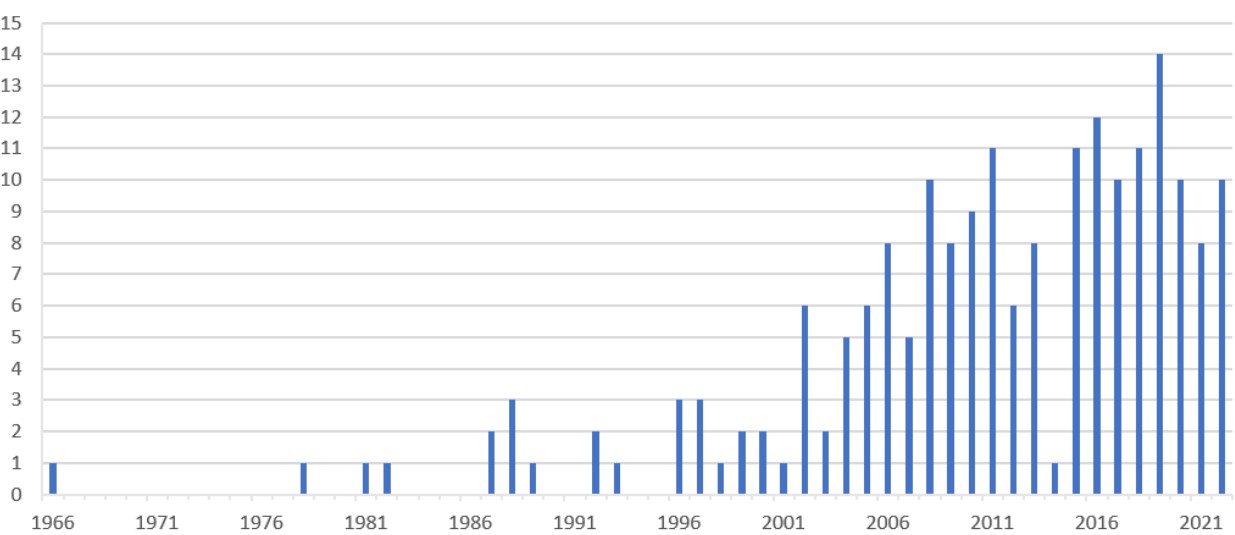

**Figure 1.** Corporate charitable giving publications between 1966 and 2021 [1–192].

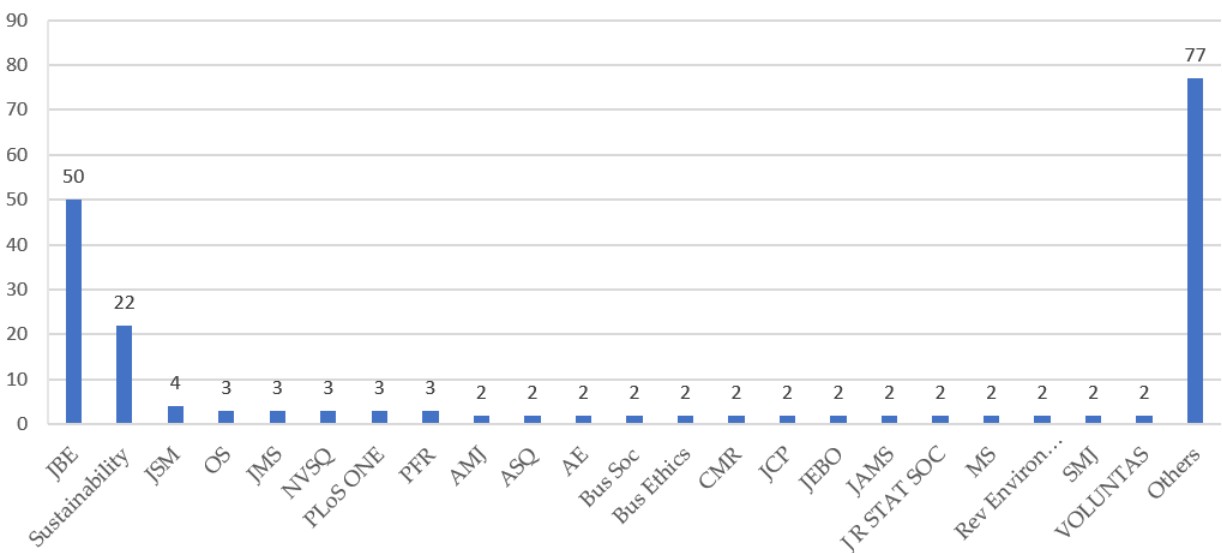

**Figure 2.** Major academic journals of corporate-charitable-giving papers [1–192].

As major players in economic activities and technological innovation, enterprises are obviously under pressure from competition. Through collating and summarizing the relevant literature, we find that corporate charitable giving has a significant impact on corporate innovation. In order to gain a better grasp of the influence path of corporate charitable giving on corporate innovation, on the basis of summarizing the existing literature, we propose how corporate charitable giving affects the path and mechanism of corporate innovation. As a first step, we summarized the specific distribution of papers in the innovation area in Table 2. A majority of the existing literature found a positive correlation between corporate charitable giving and innovation performance. Based on this, the next section presents a comprehensive analysis and review of the theoretical development of corporate charitable giving, the mediating variable of corporate charitable giving affecting corporate innovation, and then reveals the multi-level mechanisms and paths of charitable giving influencing innovation from different perspectives.

**Table 2.** Distribution papers in innovation fields.

| Author(s) | Technique and Approach | Key Findings |
|---|---|---|
| Bereskin et al. (2016) [4] | Empirical Analysis | Direct-giving activities are positively associated both with higher levels of innovation and innovation that is more influential, collaborative, and original. |
| Zhang et al. (2016) [3] | Empirical Analysis | Charitable giving has a significant inverted U-curve predictive effect on innovation performances. |
| Bereskin and Hsu (2016) [5] | Empirical Analysis | Direct giving is associated with obtaining patents that have the potential to expand the companies' expertise and range of investment opportunities. |
| Jiang et al. (2018) [17] | Empirical Analysis | There is an obviously positive correlation between philanthropic donations and innovation performance, which will be affected by the scale of the enterprise. |
| Chen and Zhou (2018) [18] | Empirical Analysis | There is a significant positive relationship between the amount of corporate charitable giving and the innovative-output performance of the firm. |
| Ou et al. (2021) [19] | Empirical Analysis | Media attention from charitable giving promotes corporate innovation in terms of both innovation intentions and innovation resources, by strengthening long-term strategic orientation for corporate development and attracting more technological resources. |

## 3. Theoretical Development of Corporate Charitable Giving

Corporate philanthropy as a business practice has been on the rise since the early 1950s, when companies mainly made cash donations to nonprofit organizations through foundations, as an effective way to demonstrate their social responsibility (Eells, 1958; Honer, 1955) [12,13]. In the 1970s and 1980s, globalization led to a trend of increased corporate mobility, which had a negative impact on the long-term relationship between companies and the workforce. Companies begin to explore the use of corporate philanthropy to restore these labor relations (Kasper and Fulton, 2006) [14]. Since the early 1990s, companies have turned to charitable giving as a response to stakeholder demands for corporate social engagement (Marinetto, 1999) [15], and to using philanthropic initiatives to advance business interests through strategic alliances with marketing, government affairs, research and development, and human-resources functions (Porter and Kramer, 2002) [16]. Over the past few years, companies have increasingly turned to corporate philanthropy as a valuable strategy to increase their competitive advantage.

With the growing interest in corporate charitable giving among industry and academia, there is a substantial body of literature on corporate philanthropy motivation. A number of studies on the motivation of charitable giving and its impact on financial performance have yielded rich results (Porter and Kramer, 2002; J. L. Campbell, 2007; Godfrey, 2005) [16,20,21]. In the complex and changing economic environment, the improvement of innovation performance is a manifestation of the enhanced technological strength of enterprises. Due to the scarcity of resources and the limitation of technology, the resources needed for enterprise innovation often cannot be obtained on their own, so external resource supplement and policy support is crucial. The essence of corporate profit-maximization determines that the motivation of corporate charitable giving is to obtain resource compensation, reduce government regulation and obtain better political resources. However, most of the existing literature always focuses on the determinants of corporate charitable giving and the motivations for giving, while few scholars relate the impact of charitable giving to the associated economic consequences. Whether charitable giving occurs out of altruism or strategic decision-making, it can lead to a mutual exchange of resources that can boost innovation. A general overview of research on corporate-charitable-giving motives is shown in Table 3.

Table 3. A General Review of Research on Corporate-Charitable-Giving Motivations.

| Research Theme: Concept | |
|---|---|
| **Level of Analysis** | **Key Ideas** |
| The altruism of corporate charitable giving | Altruistic motives are the main reason for corporate charitable giving. (Cowton 1987) [22] <br> The altruistic model of corporate philanthropy is considered to be a non-strategic interpretation of corporate giving. Companies use social standards as the basis for correct, good and just social action. Firms engage in altruistic philanthropy with the single goal of helping others. (Useem 1988; Sharfman 1994) [23,24] <br> One possible motivation is that a person actually prefers the public benefit provided by the donation. In this case, the money belonging to the charity has a utility (or reward value) that is independent of the individual's contribution. This preference for public goods can be considered "altruism", and if this is the only motivation for the donation, the behavior is called "pure altruism". (Andreoni 1989) [25] <br> There is an ethical element to the decision to engage in corporate philanthropy that cannot be ignored. (Shaw and Post 1993) [26] <br> Despite its noble goals, the altruistic model itself is often not a strong explanation for corporate philanthropy, even in the most diverse societies, which is because it ignores the profit maximization goals and other strategic objectives of corporations. (Neiheisel 1994) [27] <br> Corporate philanthropy can be driven by factors such as the sense of aesthetic pleasure. (File and Prince 1998) [28] <br> The study shows that firms who have a history of giving to charity cite altruistic motives for their behavior. (Campbell et al., 1999) [29] <br> Pro-social behavior theorists claim that managers' behavior is also motivated by ethical norms, which is a strong reason for corporate giving. (Valor 2006) [30] <br> Corporate philanthropy may be the result of top management's values of benevolence and integrity. (Choi and Wang 2007) [31] |
| The strategic nature of corporate charitable giving | Many companies which have a strong sense of corporate social responsibility, however, are turning away from traditional giving and toward a more market-driven strategic management, bottom-line approach to philanthropy. (Mescon and Tilson 1987) [32] |
| The strategic nature of corporate charitable giving | Profit maximization is an important motivation driving corporate charitable giving. (Navarro 1988) [33] <br> Managers use corporate philanthropy to promote management and corporate interests. (Haley 1991) [34] <br> From a strategic perspective, corporate philanthropy can be further divided into economic or political dimensions. (Neiheisel 1994; Young and Burlingame 1996) [9,27] <br> When companies engage in philanthropy in a more strategic and professional way, they can not only do work that has a greater impact on society, but they can also create tangible and intangible benefits for themselves in terms of goodwill, employee morale and public support. (Collins 1995) [35] <br> Firms engage in philanthropic activities as a means of improving the financial performance of their organizations. (Sanchez 2000) [36] <br> While corporate violations of environmental and labor regulations can reduce their public image, charitable giving can reduce the extent of the decline. (Williams and Barrett 2000) [37] <br> Corporate philanthropy has a positive impact on corporate financial performance because decisions about charitable giving can strategically enhance a firm's image and reputation (Porter and Kramer 2002; Godfrey 2005) [16,21] <br> Firms are becoming increasingly strategic in their philanthropic activities. (Saiia et al., 2003) [38] <br> Some companies are increasingly emphasizing corporate charitable giving as a strategic tool. (Patten 2008) [39] <br> High social sentiment and politically-related economic motivations should trigger a more efficient market reaction to corporate philanthropic involvement, which could then address more directly and effectively the strategic motivation and the "insurance-like" protection property of reputational capital behind corporate philanthropy. (Gao et al., 2012) [40] <br> Managed rent withdrawals are motivated as corporate donations. (Masulis and Reza 2015) [41] <br> Corporate reputation effects emphasize on altruistic motivation, while strategic philanthropy focuses more on egoism. (Xu et al., 2017) [42] <br> Corporate decisions to be charitable are not based solely on purely philanthropic motives, but depend more on the source of revenue. (Oh et al., 2018) [43] <br> When controlling shareholders intend to transfer wealth out of the company, they may use the company's resources for charitable giving. (Chen et al., 2018) [44] |

### 3.1. The Altruism of Corporate Charitable Giving

The altruistic model of corporate philanthropy (Sharfman, 1994; Useem, 1988) [23,24] is considered to be a non-strategic interpretation of corporate giving. According to this theory, companies use social standards as the basis for correct, good and just social action. Firms engage in altruistic philanthropy with the single goal of helping others, so philanthropy is considered independent of the operational pressures that generate profits. As previous researchers have pointed out (Shaw and Post, 1993) [26], there is an ethical element to the decision to engage in corporate philanthropy that cannot be ignored. Previous research has found that corporate philanthropy can be driven by factors such as the sense of aesthetic pleasure (File and Prince, 1998) [28] or altruism (Campbell et al., 1999) [29]. Firms have an obligation to engage in corporate philanthropy, rather than out of any self-interested financial considerations (Shaw and Post, 1993) [26]. In sum, there is a substantial body of literature that reveals the decision to engage in philanthropy is driven by moral obligation.

As Sharfman (1994) [24] has pointed out in the altruistic model, the essence of philanthropy is ethical. Thus, managers have an ethical responsibility to allocate the firm's resources in a way that promotes the overall welfare of society, regardless of whether these actions lead to specific results, such as increased profits or an enhanced image. Choi and Wang (2007) [31] asserted that corporate philanthropy may be the result of top management's values of benevolence and integrity. In addition to economic reasons, pro-social behavior theorists claim that managers' behavior is also motivated by ethical norms, which is a strong reason for corporate giving (Valor, 2006) [30]. From this perspective, the motivation for corporate charitable giving can be categorized as altruistic. Despite its noble goals, the altruistic model itself is often not a strong explanation for corporate philanthropy, even in the most diverse societies, because it ignores the profit-maximization goals and other strategic objectives of corporations (Neiheisel, 1994) [27].

### 3.2. The Strategic Nature of Corporate Charitable Giving

At the level of strategic nature, corporate philanthropy is strategically motivated by the intent to contribute to direct monetary benefits, in the same way as any other corporate function. From a strategic perspective, corporate philanthropy can be further divided into economic or political dimensions (Neiheisel, 1994; Young and Burlingame, 1996) [9,27]. The economics of strategic philanthropy suggests that firms engage in philanthropic activities as a means of improving the financial performance of their organizations (Sanchez, 2000) [36], while the political view is that firms engage in corporate philanthropy because of the political and institutional pressure exerted on them by key environmental actors (Neiheisel, 1994) [27].

Based on the previous research, the strategic motivation of corporate philanthropy can be explained from the following two perspectives. Firstly, increased global competition requires firms to build a competitive advantage through various channels. Corporate philanthropy can help firms gain brand recognition and loyalty, promoting themselves as "socially responsible" firms. Secondly, the elimination of government agencies and cuts to state budgets that previously supported the arts and social services have stimulated the growth of voluntary agencies and private foundations. As government support dwindles, a growing number of private voluntary organizations are trying to raise money from private firms. In turn, these firms set up foundations to pass on these requests.

In the research on corporate charitable giving, a number of studies have focused on the impact of charitable giving on corporate value, financial performance, and other factors as a means of enhancing corporate value. When charitable giving is taken as part of marketing related to firms, it is usually directly related to the improvement of corporate performance. Some scholars argue that corporate philanthropy has a positive impact on corporate financial performance because decisions about charitable giving can strategically enhance a firm's image and reputation (Godfrey, 2005; Porter and Kramer, 2002) [16,21]. At the same time, corporate charitable giving often serves as a means for firms to reduce

the risks associated with resource acquisition from the perspective of resource dependence (Haley, 1991) [34].

Furthermore, firms also try to create a positive social image through charitable giving to mitigate or offset negative social impact in other areas. For instance, Williams and Barrett (2000) [37] examined the impact of corporate-charitable-giving programs on the association between the number of corporate violations of EPA (Environmental Protection Agency) and OSHA (Occupational Safety and Health Administration) regulations and their public image, and found that while corporate violations of environmental and labor regulations can reduce their public image, charitable giving can reduce the extent of the decline. Therefore, contrary to the CSR perspective (Carroll, 1991) [45], firms may increase charitable giving to maintain their positive social image, rather than devote resources to correcting negative social impact in other areas.

## 4. Perspectives on Corporate Charitable Giving and Enterprise Innovation

Whether or not the motivation for corporate philanthropy is defined as an altruistic model, a commitment to the common good or community investment or marketing, research demonstrates that giving ultimately benefits firms. In addition, firms with robust corporate-charitable-giving programs perform better than others. However, there is still a lack of comprehensive and systematic research on the perspective from which corporate charitable giving affects innovation. By consulting the following literature on the consequences of corporate charitable giving and the antecedents of enterprise innovation, this paper extracts the impact perspectives of the following aspects from the research, as shown in Table 4. Due to the resource supply and external support from stakeholders of enterprise-innovation activities, firms can use charitable giving programs to outsource R&D under the resource-dependence theory. By establishing links between firms and research organizations, firms will be able to gain access to a wider range of knowledge and exploration opportunities, as well as contribute significantly to the development of corporate research-networks and enterprise innovation (Bereskin et al., 2016) [5]. For innovative firms that need government support or subsidies for innovative activities, it is critical to develop positive relationships with regulators. Moreover, corporate charitable giving can improve the ability of firms to interact positively with government regulators, and enhance their political reputation and influence in society. In addition, increased media attention on corporate charitable giving can help reduce information asymmetry and build a positive image of social responsibility (Du et al., 2010) [46]. Through the signal effect of public giving, firms can convey a positive image of corporate social responsibility to stakeholders, and promote their innovative activities. Moreover, within the framework of the social-responsibility exchange theory, firms can increase social resources by charitable giving, and thus improve the competitiveness of the firms (Porter and Kramer, 2002) [16]. From the diverse studies into the above perspectives, this paper summarizes the impact perspective into four aspects: research networks, political reputation, media attention, and resource allocation, and then describes the mechanism and approaches to the impact of charitable giving on innovation, in detail.

**Table 4.** Four categories of research from different perspectives.

| Category | Author(s) | Technique and Approach | Conclusion |
|---|---|---|---|
| Technological Network | Bereskin and Hsu (2016) [5] | Empirical Analysis | Direct giving is associated with obtaining patents that have the potential to expand the companies' expertise and range of investment opportunities. |
| | Wang (2011) [6] | Empirical Analysis | The positive philanthropy–performance relationship is stronger for firms with greater public visibility and for those with better past performance. |

**Table 4.** *Cont.*

| Category | Author(s) | Technique and Approach | Conclusion |
|---|---|---|---|
| Political Reputation | Godfrey (2005) [21] | Literature Review | Corporate philanthropy can generate positive moral capital among communities and stakeholders. Moral capital can provide shareholders with insurance-like protection for a firm's relationship-based intangible assets. |
| | Chen et al. (2018) [44] | Empirical Analysis | The negative association between philanthropy and tunneling is stronger when firms are faced with more severe agency conflicts, as indicated by lower largest-shareholding, fewer growth opportunities, lower state-ownership, and weaker product-market competition. |
| | Campbell and Slack (2007) [20] | Empirical Analysis | Individual societies privilege local (to themselves) causes over others when it comes to the distribution of charitable and community donations. |
| | Bertrand et al. (2020) [47] | Empirical Analysis | In the absence of disclosure requirements, charitable giving may be a form of corporate political-influence undetected by voters and subsidized by taxpayers. |
| | Li et al. (2015) [48] | Empirical Analysis | A significant and positive relationship between political connections and the likelihood and extent of firm contributions. A stronger relationship between political connections and corporate philanthropy in non-state-owned firms. |
| | Zhang et al. (2016) [49] | Empirical Analysis | Firms whose executives have ascribed bureaucratic connections are more likely to use their connections as a buffer from governmental donation-pressure. |
| | Zhang et al. (2013) [50] | Empirical Analysis | The government–enterprise bonding effect of charitable giving is more pronounced among state-owned corporations than among non-state-owned corporations. |
| Media Attention | Jia and Zhang (2018) [51] | Empirical Analysis | Firms that make generous donations at the beginning of a new city secretary's tenure receive more attention from representatives of new local leaders, especially firms that were politically disadvantaged under a predecessor's governance. |
| | Ou et al. (2021) [19] | Empirical Analysis | Media attention from charitable giving promotes corporate innovation in terms of both innovation intentions and innovation resources, by strengthening long-term strategic orientation for corporate development and attracting more technological resources. |
| | Jia and Zhang (2015) [52] | Empirical Analysis | Corporations that are highly visible in the news media are more likely to engage in CPR and donate more money. |
| Resource Regulation | Chen and Zhou (2018) [18] | Empirical Analysis | There is a significant positive relationship between the amount of corporate charitable giving and the innovative-output performance of the firm. |
| | Cha (2021) [11] | Literature Review | The underlying mechanisms of corporate philanthropy–firm performance relationship. |

### 4.1. Perspective of Technological Network

In an increasingly competitive marketplace, relevant resources and technical support from stakeholders can effectively facilitate the innovation activities of firms. At the institutional level of research, Bereskin et al. (2016) [4] confirmed through research that there is a significant positive correlation between direct giving and collaborative innovation, which suggests that firms with more direct donations learn more from nonprofits and have broader innovation networks, i.e., direct giving activities are positively associated with higher levels of innovation and more impactful, collaborative, and original innovation. In addition, the impact of direct giving on innovation is more pronounced in opaque firms and in more innovative, competitive industries. By leveraging the capabilities of nonprofits, much of what is disclosed as direct giving enables firms to explore new technologies

and expand their expertise. In addition to the research on the correlation between direct giving and collaborative innovation, an empirical study of pharmaceutical companies by Bereskin and Hsu (2016) [5] also shows that direct donations can be used quite successfully by pharmaceutical companies to foster relationships with academia, increasing their innovation capacity. Their findings suggest that direct giving is associated with higher quantity and quality of innovation, more impactful patents, and more productive R&D spending. Furthermore, these results illustrate the different motivations for firms to use direct giving types of philanthropy to develop innovative products in collaboration with research partners. As for the research on the antecedents of enterprise innovation, Ritter and Gemünden (2003) [53] believe that network competence has a strong positive influence on the extent of interorganizational technological collaborations, and on a firm's product- and process-innovation success. External focus, decentralization, and IT are also associated with improved product-innovation capabilities (Tambe et al., 2012) [54].

### 4.2. Perspective of Political Reputation

The reputation of a firm affects not only its access to political resources, but also the resources the government provides in the form of innovation-policy protections and innovation-development resources. In China, corporate philanthropy is an informal mechanism for private firms to develop and safeguard their activities. Numerous studies have found the political benefits of corporate philanthropy, including helping firms gain political legitimacy and allowing them to access valuable political resources that are critical to long-term survival and financial success (Hillman, 2005) [55]. Corporate philanthropy is particularly valued when governments do not have sufficient resources to engage in community and social-welfare programs, as corporate charitable giving can help ease their burden. According to Godfrey (2005) [21], corporate philanthropic reputation helps to protect a firm's relationship with its stakeholders, thereby reducing the risk of losing critical resources. At the same time, strengthening the reputation of a firm can serve to signal the government's recognition of the firm, as well as to strengthen the stakeholder's relationship with the firm. Thus, it reduces the risk perception of collaborators towards the firm, helps the firm to obtain the production resources and innovation factors needed for innovation, and has a positive role in promoting the innovation activities of firms. In a similar way, Fuertes-Callén and Cuéllar-Fernández (2014) [56] argue that commercialization and reputation are key elements in product success. Another study shows that there is a direct relationship of statistical significance between the level of enhancing the perceived organizational-reputation and achieving strategic creativity (Keshta et al., 2020) [57].

### 4.3. Perspective of Media Attention

As an efficient way to fulfill corporate social responsibility, charitable giving helps firms shape their corporate image for long-term sustainable development. In fact, it reflects the efforts of firms to be socially responsible. The news media, on the other hand, plays a role in collecting, processing and disseminating information on the process of implementing charitable giving by firms. In addition, the agenda-setting effect of the news media draws the attention of stakeholders to corporations (Jia and Zhang, 2018) [51]. Owing to the powerful influence on what people think which is set by the media news-coverage and agenda, issues or firms that are widely covered by the mass media receive more attention, so that it is easier to push firms to invest more resources in sustainable economic activities such as technological innovation and environmental protection, with the attention of many stakeholders. In terms of the relationship between media attention and enterprise innovation, Flipse and Osseweijer (2014) [58] note that the quick response-speed of media promotes innovative practices. Mount and Martinez (2014) [59] find that collective intelligence emerges from the many-to-many interactions supported by social media during open-innovation activities. In additionsocial media is seen as an enabler and driver of innovation (Bhimani et al., 2019) [60]. As discussed previously, Ou Jinwen et al. (2021) [19] show that media attention to charitable giving can contribute significantly to the innovation

performance of firms, by promoting a longer-term strategic direction of development for the firm, and that attracting more technological resources, i.e., media attention, to charitable giving, can contribute significantly to a firm's innovation performance.

### 4.4. Perspective of Resource Regulation

Various inputs are necessary for firms to innovate technologically, which is the foundation of enterprise innovation. Despite this, firms do not have access to all the resources needed for innovation, and they cannot obtain optimal innovation-performance without investing their own resources; the high cost of acquiring innovation resources and poor corporate management prevent a firm from making optimal use of these resources. Hence, internal, redundant resources and existing assets, which refer to slack resources within the company, can often be used as a resource supply-channel for corporate development and innovation. Slack resources are resources that are kept within the firms beyond the actual operational needs of the firms (Bougeois, 1981) [61]. Several studies have demonstrated that redundant resources and the efficiency of corporate-asset utilization have a very important moderating effect on enterprise innovation. In turn, charitable-giving behavior also happens to be influenced by organizational resource-characteristics. For example, Collis' (1991) study [62] categorizes one of the roles of redundant resources as supporting intra-firm innovation activities by providing a more realistic route to supply resources for enterprise-innovation activities. Furthermore, based on stakeholder theory and resource dependence theory, the study by Chen and Zhou (2018) [18] demonstrated a significant positive relationship between the amount of corporate charitable giving and the innovative-output performance of firms. Moreover, when firms have more redundant resources and higher asset-utilization efficiency, i.e., higher profitability, the contribution of firms' donation behavior to firms' innovation output is stronger. Thus, resource regulation plays an essential moderating role in the influence mechanism of corporate charitable giving and enterprise-innovation performance. Without loss of generality, Tan and Peng (2003) [63] note that the impact of different types of organizational slack on the innovation activities of enterprises differs significantly, depending on their individual characteristics. More precisely, Morrison et al. (2000) [64] found that redundant resources in firms can help encourage an environment which is conducive to innovation, and thus drive innovation output in firms. Geiger and Cashen (2002) [65] believe that Internal redundancy may become a resource supply-channel for firms to implement innovation projects. Table 5 below represents four different perspectives on the impact of innovation.

**Table 5.** Four different types of perspectives on the impact of innovation.

| Category | Author(s) | Technique and Approach | Conclusion |
|---|---|---|---|
| Technological Network | Ritter and Gemünden (2003) [53] | Empirical Analysis | Network competence has a strong positive influence on the extent of interorganizational technological-collaborations and on a firm's product- and process-innovation success. |
| | Tambe et al. (2012) [54] | Empirical Analysis | External focus, decentralization, and IT are associated with improved product-innovation capabilities. |
| Political Reputation | Fuertes-Callén and Cuéllar-Fernández (2014) [56] | Empirical Analysis | Commercialization and reputation of innovative product intervene in the relationship between innovation and product success. |
| | Keshta et al. (2020) [57] | Empirical Analysis | There is a direct relationship of statistical significance between the level of enhancing the perceived organizational reputation and achieving strategic creativity. |

**Table 5.** *Cont.*

| Category | Author(s) | Technique and Approach | Conclusion |
|---|---|---|---|
| Media Attention | Flipse and Osseweijer (2013) [58] | Empirical Analysis | The quick response-speed of media promotes innovative practices. |
| | Mount and Martinez (2014) [59] | Cases Overview | Collective intelligence emerges from the many-to-many interactions supported by social media during open-innovation activities. |
| | Bhimani et al. (2019) [60] | Literature Review | Social media is seen as enabler and driver of innovation. |
| Resource Regulation | Morrison et al. (2000) [64] | Empirical Analysis | Redundant resources in firms can help encourage an environment conducive to innovation and thus drive innovation-output in firms. |
| Resource Regulation | Geiger and Cashen (2002) [65] | Empirical Analysis | Internal redundancy may become a resource supply-channel for firms to implement innovation projects. |
| | Tan and Peng (2003) [63] | Empirical Analysis | The impact of different types of organizational slack on the innovation activities of enterprises differs significantly, depending on their individual characteristics. |

*4.5. Theoretical Basis and Potential Mechanisms*

Based on the literature review of the above impact perspectives, this research identified the fact that there have been studies on how corporate charitable giving affects enterprise-innovation performance from four theoretical perspectives: research network, political reputation, media attention and resource adjustment. In sum, the following section of this paper will provide a summation of the analytical process and multi-level model under this adjustment mechanism by combining the preconditions, key factors, and adjustment mechanism in the process of corporate charitable giving which affects innovation, and delving into each theoretical perspective, to fill an important theoretical gap in this field. By extracting the development path in the real situation, this paper first summarizes the prerequisites for corporate charitable giving within two categories: altruistic and strategic, which are the antecedents to corporate charitable giving and the focus of much corporate-giving research. Using diverse motives, companies fulfill social responsibilities through charitable donations, acquire technical support, political resources, or other resources that contribute to their development, and influence their innovation performance through key factors under the four perspectives outlined above, thus creating a path system of perfect mechanisms to influence their innovation performance, as depicted in Figure 3.

The influence path in our multi-level model indicates that altruistic motivation or strategic motivation drives firms to implement charitable giving in the first place, which is one of the prerequisites of our model. After this, corporate charitable giving enhances enterprise-innovation performance by promoting three key factors: broadening a firm's research network, gaining political reputation, and increasing media-attention, which explains the left "+". In addition, based on resource dependence theory and the important influence of profitability on innovation resources and firm competitiveness, the model treats redundant resources and profitability from the resource-regulation perspective as moderating variables in the positive relationship between charitable giving and enterprise innovation, and indicates that redundant resources and profitability can play positive moderating roles in the positive relationship, which represents the right "+".

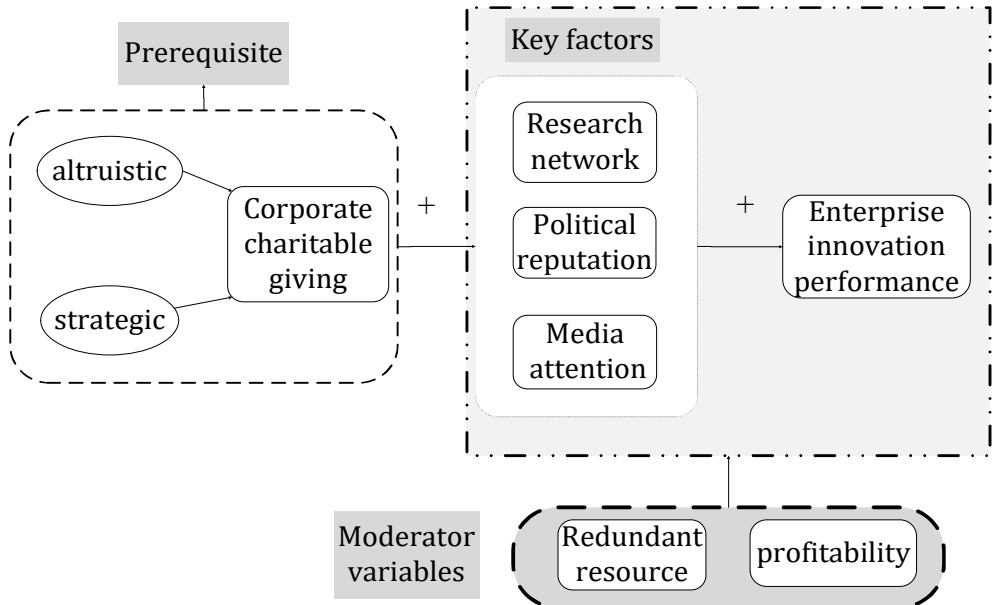

**Figure 3.** A multi-level model of the impact of corporate charitable giving on enterprise innovation.

## 5. Conclusions and Discussion

### 5.1. Conclusions

Over the past half-century, corporate philanthropy and charitable giving have been studied across and between disciplines. Based on the literature review on corporate charitable giving and enterprise innovation, this paper classifies and elaborates on the development of theoretical perspectives and the differentiation of research themes in this field, based on an inductive analysis. We elaborate the mechanism and pathways through which charitable giving influences enterprise innovation from the perspective of four influences: research networks, political reputation, media attention, and resource adjustment. In addition, we summarize the moderating role of external factors such as macro-level and meso-level. The altruistic and strategic motivations of corporate philanthropy provide the theoretical foundations for analyzing how charitable giving is related to innovations in business; the multi-level path model is the internal logic that explains how charitable giving is related to enterprise-innovation performance, firms' redundant resources and profitability provide positive incentives for organizational-innovation performance, while government policies on charitable giving ultimately shape firms' innovation models and instruments. Due to the limitations of existing studies in terms of models, variables and mechanisms, we summarize and expound the significance of the path model based on literature review, as follows.

On the one hand, the summary of the results of the practice and theory of the corporate-charitable-giving industry summarizes the differentiation paths and research hotspots in the field of charitable-giving research in a more intuitive way, which helps subsequent studies to grasp more precisely the theoretical basis and research background of corporate charitable giving and enterprise innovation in practice. On the other hand, starting from a multi-level analysis model with prerequisites, key factors, and moderating variables, we summarize the objective conditions, driving forces, and internal connections which enable charitable giving to function and refine systematic influence-mechanisms from four perspectives: research network, political reputation, media attention, and resource adjustment. Despite the efforts to identify much of the literature, the relevant literature in this field has not yet formed a systematic theoretical system on the influence mechanisms of corporate charitable giving and enterprise innovation, which is not conducive to the communication and interoperability of related studies. Most of the literature focuses on the interpretation of the influence of a single factor on the correlation, but fails to fully provide a comprehensive explanation of the influence mechanism. This paper reveals the

fundamental mechanism and influence path of charitable giving on innovation by linking corporate charitable giving and enterprise innovation, and fills a gap in the research on the relationship between charity and enterprise innovation, through a multidisciplinary framework. By providing a systematic perspective and theoretical foundation, this research helps to significantly improve the depth and breadth of future research in the field. In addition, this paper presents businesses with invaluable suggestions about how to improve their enterprise-innovation performance with corporate charitable giving, and through which specific paths.

*5.2. Discussion*

After summarizing the influence mechanism, evolutionary path, and multi-level model, we believe that future research can be further extended in the following areas.

First of all, there are limitations in previous studies on the measurement of charitable-giving behavior. The criteria for charitable giving are multidimensional, coming not only in the form of direct giving and foundation giving, but also in the form of strategic and altruistic motivations. Similarly, strategic motives can be divided into various types, such as corporate financial-performance, reputational performance, and political legitimacy (Cha, 2021) [11]. However, most of the current literature only focuses on single cash-donation, without considering the motivation of charitable giving and the comprehensive impact of different forms of enterprise-innovation performance. In the future, we can broaden the research paradigm of charitable giving by including variables such as motivation to give in the research model, to expand the coverage of the research population and thus examine the diversity of the impact effects of charitable giving, more comprehensively.

Secondly, the existing research is deficient in the multidimensional examination of the impact of charitable giving on innovation. For the present, most of the literature has examined the mediating role of only one key factor in the influence mechanisms, without considering the combined effects of other key factors or moderating variables. In essence, the comprehensive effect of different variables will greatly affect the design of the innovation mechanism and the accomplishment of innovation performance. Therefore, future research still needs to broaden the horizon and consider the comprehensive role of multiple key factors in the mechanism. Specifically, this part includes what changes will take place in the role of key factors under compound influence, at what level the effects of the two factors can cancel each other out, and what measures can be taken to alleviate this role. Through multi-level research, stakeholders and investors can then target the implementation of measures to promote enterprise innovation and avoid potential problems.

Finally, future research needs to further deepen the measurement of firm innovation as well. Present studies on the measurement of enterprise innovation basically consider only the traditional indicators, such as the amount of innovation investment of a firm as a measure of the level of R&D, and lack research into and measurement of other qualitative variables, such as innovation patterns and innovation efficiency. However, the level of enterprise-innovation outcomes is often attributed to the choice of innovation strategies and the efficiency of innovation quality. As a result, future research should further refine the measurement perspective of enterprise-innovation performance. To fill this momentous research gap in the measurement of enterprise innovation, and to expand future research ideas on enterprise innovation, future research could measure the role of charitable giving in influencing enterprise innovation, widening the scope to become more comprehensive and complete.

**Author Contributions:** Conceptualization, X.G. and X.S.; Methodology, X.G. and Y.L.; Literature Summary, L.X. and J.J.; Literature Analysis, L.X. and Y.L.; Conclusions, L.X. and X.G. All authors have read and agreed to the published version of the manuscript.

**Funding:** This research was supported in part by NSFC special supporting funding of CAUC (Grant No. 3122022PT08), SAFEA High-End Foreign Experts Project (Grant No. G2022202001L).

**Conflicts of Interest:** The authors declare no conflict of interest.

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
