# Peer review of "How Does Corporate Charitable Giving Affect Enterprise Innovation? A Literature Review and Research Directions"

_sustainability, doi:10.3390/su142315603_

Round 1

Reviewer 1 Report (Previous Reviewer 2)

The authors have improved the manuscript but few comments still need to be addressed. e.g, the distribution of journals and details of studies used for the literature review (n~196) as shown in Table 1. But the figure 2 sum of studies journal wise count is 187. However, the complete references at the end  show only 65 studies. Please verify and validate your argument and just provide the exact number of papers reviewed. If you want to keep the 65 references  the Table 1 and Figure 2 numbers should be modified accordingly. Or if want to keep same number of table 1  (196), the figure 2 and final references must the the same. Please clarify and correct it  

Author Response

Reviewer 2 Report (Previous Reviewer 1)

The authors have done an extensive revision based on the previous comments (the manuscript ID 1905962). I have a few comments to improve the flow of reading in the following.

1. Please re-check Section 2. Specifically, Section 2 is 'Methodology'. It talked about how the authors conducted the systematic review, how to get data, etc. However, Section 2.2 is your findings which should not be in this Section. Please revise.

2. The authors mentioned that they applied the process of systematic review by Bekkers and Wiepking. Can the author indicate the search terms used in this study? How many articles obtained from the first stage? What are the criteria to remove unrelated articles to get the final articles of 196?

3. Did the authors search articles from different databases (e.g., ScienceDirect, ProQuest, etc.)? The authors need to ensure that the search terms and other criteria are the same for all databases.

Round 2

Reviewer 2 Report (Previous Reviewer 1)

I see that the authors have made all the modifications and add the comments. It is ok for publication. Congratulations!

This manuscript is a resubmission of an earlier submission. The following is a list of the peer review reports and author responses from that submission.

Round 1

Reviewer 1 Report

I enjoyed reading your paper and this topic sounds interesting and can be potential of publication. Both major and minor issues for improvement are provided. Before resubmitting, please consider the following:

1.      In the Introduction, the authors mentioned that there were several studies on corporate charitable giving to innovation. Please add the gap or the necessity to conduct this review literature.  

2.      What are your research questions (RQs)?  The authors need to develop RQs and explain why the study is needed and what are the justifications for undertaking this study.

3.      For data collection and analysis, the authors have to inform readers on how to collect the 196 papers. Are there any criteria for selecting? For instance, the authors may consider applying PRISMA protocol (https://www.prisma-statement.org//PRISMAStatement/FlowDiagram) or others.

4.      What are the methods/ techniques that the authors used for the analysis?

5.      In Section 3, how did the authors clarify the corporate charitable giving into 4 themes? The authors may need to elaborate the review among 196 papers and show the impact on corporate charitable giving and enterprise innovation of each theme. In other words, what are the contributions of the studies for each theme?

6.      Please check the error in Line 291.

7.      In Line 319, it shows Figure 1; however, there is no ‘Figure 1’ mentioned in the main text.

8.      The authors need to add contributions to theory and literature explaining how this research fills or advances the existing theories. Especially, the authors mentioned that they proposed the novel theoretical perspective.

9.      Referred to no.8, in Figure 1, the authors should explain how did you construct this model. Moreover, how did you explain a ‘+’ sign?

10.  The authors need to add managerial implication explaining how managers can use your research findings in businesses.

11.  Please check the journal requirement for manuscript structure such as reference format.

Overall, you have a good research approach. Following above suggestions would overcome deficiencies of your manuscript.

Author Response

Response to Reviewer 1 Comments

I enjoyed reading your paper and this topic sounds interesting and can be potential of publication. Both major and minor issues for improvement are provided. Before resubmitting, please consider the following:

Point 1: In the Introduction, the authors mentioned that there were several studies on corporate charitable giving to innovation. Please add the gap or the necessity to conduct this review literature.  

Response 1:

Thank you for your advice, I have added the gap in the introduction part.

Point 2:What are your research questions (RQs)?  The authors need to develop RQs and explain why the study is needed and what are the justifications for undertaking this study.

Response 2:

Our research questions is “how does corporate charitable giving affect enterprise innovation?”

By reviewing articles on how corporate charitable giving affects corporate innovation, a theoretical influence mechanism is constructed to provide a theoretical basis for later research.

Point 3:For data collection and analysis, the authors have to inform readers on how to collect the 196 papers. Are there any criteria for selecting? For instance, the authors may consider applying PRISMA protocol (https://www.prisma-statement.org//PRISMAStatement/FlowDiagram) or others.

Response 3:

To be the very first step, we used data collection to delineate the boundaries of our literature review. Data collection of papers and documents was published between 1950 and 2021. First, we searched major business database, such as ScienceDirect, ProQuest, Springer Link, Emerald and so on. Corporate charitable giving literature were looked at, conducted on common terms and variations of words associated with related words. Second, we refine the research scope of the literature and divide all the literature into seven research topics according to different types of research, which are Corporate Charitable Giving and Performance, Corporate Charitable Giving and Innovation, Corporate Charitable Giving and Corporate Social Responsibility, Factors influencing corporate charitable giving, The Impact of Corporate Charitable Giving on Business, A few literature reviews related to corporate charitable giving and others. The proportion of different research contents is shown in Figure1.

Figure1. Classification of research related to corporate charitable giving

Data analysis

As for corporate donation, previous studies have paid more attention to the effectiveness of corporate donation and the different effects of corporate donation on consumer behavior. Early researchers (Fry et al., 1982) [1] found that corporate philanthropy provides direct benefits to participants and benefits to individuals, organizations and communities in the external business environment.

Although there is relatively little direct evidence in corporate charity literature, a large number of marketing studies related to causes show that the contribution of enterprises has a significant positive impact on consumers' perception. Specifically, when a company donates more money to charity, consumers will have a more positive evaluation of the company. For example, Pracejus et al. (2003) [2]found that the donation amount has a positive impact on product selection when the price is constant. Similarly, the research of Haruvy et al. (2009) [3]shows that the donation amount of charity auction (expressed as a percentage of auction income) has a positive impact on auction income.

In previous studies, a large number of scholars have studied donation strategies. For example, Hildebrand (2017) [4]proved in the research that the evaluation advantages given by in-kind contributions actually depend on the characteristics of CSR problems caused by contributions, such as their perceived controllability, by comparing consumers' preferences for two kinds of enterprises-monetary donations and in-kind donations, and through scenario experiments and data analysis. Jin (2018) [5] put forward two new donation strategies, focusing on the amount or frequency of corporate donations, and verified that consumers have different cognitive and behavioral responses to enterprises adopting different donation strategies. Krishna and Rajan (2009) [6] found that even in a competitive environment, charity-related promotions can improve profitability, and in fact, even for those products directly involved in promotions, customers' willingness to pay for company products can be improved.

The next section presents a comprehensive analysis and review about Industry practice and theoretical development of corporate charitable giving, and then reveals the multi-level mechanisms and paths of charitable giving influencing innovation from different perspectives.

Point 4:What are the methods/ techniques that the authors used for the analysis?

Response 4: Literature analysis method.

Point 5:In Section 3, how did the authors clarify the corporate charitable giving into 4 themes? The authors may need to elaborate the review among 196 papers and show the impact on corporate charitable giving and enterprise innovation of each theme. In other words, what are the contributions of the studies for each theme?

Response 5: The content and conclusion of each research topic are part of the intermediary variables that constitute the multi-level analysis model. This paper selects representative articles from each topic as the reference content of the model.

Point 6:Please check the error in Line 291.

Response 6: We have revised this question in our article.

Point7:In Line 319, it shows Figure 1; however, there is no ‘Figure 1’ mentioned in the main text.

Response 7: The figure 1 is at the top of line 391.

Point 8:The authors need to add contributions to theory and literature explaining how this research fills or advances the existing theories. Especially, the authors mentioned that they proposed the novel theoretical perspective.

Response 8: The paper connected corporate charitable giving with enterprise innovation, developed a multi- level and multidisciplinary framework which reveals the fundamental mechanism and the influence path of charitable giving on innovation, which fill the research gap of the theoretical mechanism between charity giving and enterprise innovation. By providing a systematic perspective and theoretical foundation, this research helps to improve the depth and breadth of great significance for future research in the field.

Point 9:Referred to no.8, in Figure 1, the authors should explain how did you construct this model. Moreover, how did you explain a ‘+’ sign?

Response 9: From the influence path in our multi-level model, altruistic motivation or strategic motivation drives firms to implement charitable giving in the first place, which is the prerequisite in the multi-level model. After that, corporate charitable giving enhances enterprise innovation performance by promoting three key factors: broadening a firm's research network, gaining political reputation, and increasing media attention, which explains the left “+”. In addition, based on the resource dependence theory and the important influence of profitability on innovation resources and firm competitiveness, the model treats redundant resources and profitability from the resource regulation perspective as moderating variables in the positive relationship between charitable giving and enterprise innovation, and indicates that redundant resources and profitability can play a positive moderating role in the positive relationship, which represent the right “+”.

Point 10:The authors need to add managerial implication explaining how managers can use your research findings in businesses.

Response 10: This paper provide businesses with an invaluable suggestion about how to improve their enterprise innovation performance with corporate charitable giving , and through which specific paths.

Point 11:Please check the journal requirement for manuscript structure such as reference format.

Overall, you have a good research approach. Following above suggestions would overcome deficiencies of your manuscript.

Response 11: Thanks for the suggestion, we have revised the reference format.

Reference:

  • Fry L W, Keim G D, Meiners R E. Corporate contributions: Altruistic or for-profit[J]? The Academy of Management Journal, 1982, 25(1):94–106.
  • Pracejus J, Deng Q, Olsen G, Messinger P. Fit in cause-related marketing: An integrative retrospective[J]. Journal of Global Scholars of Marketing Science, 2020, 30(2):105–114.
  • Haruvy E, Leszczyc P. Bidder motives in cause-related auctions[J]. International Journal of Research in Marketing, 2009, 26(4):324-331.
  • Hildebrand D, Demotta Y, Sen S, Valenzuela A. Consumer Responses to Corporate Social Responsibility (CSR) Contribution Type[J]. Journal of Consumer Research, 2017, 44(4):738-758.
  • Jin L, He Y. How the frequency and amount of corporate donations affect consumer perception and behavioral responses[J]. Journal of the Academy of Marketing Science, 2018, 46(6):1072-1088.
  • Krishna A, Rajan U. Cause Marketing: Spillover Effects of Cause-Related Products in a Product Portfolio[J]. Management Science, 55(9):1469-1485.

Reviewer 2 Report

Review Report

How does corporate charitable giving affect enterprise innovation? A literature review and research directions? A literature review and research directions

This paper reviewed the literature to clarify the relationship, the current state, and the differentiation trends of research topics and discusses the industry operations and theoretical development of corporate charitable giving. Please read any good systematic review article to see the format. 

The following corrections must be made to improve the quality of this research. 

Introduction 

 Please state the key objectives of the research study and how they can be answered, as you mentioned. The introduction should not discuss the process of review. It instead can give an overview of the research area. See line no. 52-55.  Please modify accordingly. 

Systematic Literature Review 

Have you used the Systematic literature review protocol? If yes, please summarize the significant findings in table form and show the distribution of journals and details of studies used for the literature review (n~196). However, the complete references show only 36 studies. Please verify and validate your argument. 

Methods 

Suppose authors used the Systematic literature review protocol. Please write exclusion and inclusion criteria in the methodology section.  Please clarify and justify. In the review article, we have a methodology section where we discuss how you conducted the systematic review protocol.  Include this section after the introduction.  

Data Analysis 

Please improve the quality of this section. I can’t see the tables and charts we usually do during the systematic review. You need to justify your findings and how you reviewed the papers. 

Discussion and findings 

This title is missing; there is discussion but unguided. Please revise and include a discussion section on how you concluded and came up with a proposed framework. 

Conclusion 

 This section should be at the end of your paper. You can discuss research limitations that are missing and future call after discussion.  The conclusion section should only conclude your findings

Author Response

Response to Reviewer 2 Comments

How does corporate charitable giving affect enterprise innovation? A literature review and research directions? A literature review and research directions

This paper reviewed the literature to clarify the relationship, the current state, and the differentiation trends of research topics and discusses the industry operations and theoretical development of corporate charitable giving. Please read any good systematic review article to see the format. 

The following corrections must be made to improve the quality of this research. 

Introduction 

 Please state the key objectives of the research study and how they can be answered, as you mentioned. The introduction should not discuss the process of review. It instead can give an overview of the research area. See line no. 52-55.  Please modify accordingly. 

 Response : We have revised this question in our article.

Systematic Literature Review 

Have you used the Systematic literature review protocol? If yes, please summarize the significant findings in table form and show the distribution of journals and details of studies used for the literature review (n~196). However, the complete references show only 36 studies. Please verify and validate your argument. 

  Response :

To be the very first step, we used data collection to delineate the boundaries of our literature review. Data collection of papers and documents was published between 1950 and 2021. First, we searched major business database, such as ScienceDirect, ProQuest, Springer Link, Emerald and so on. Corporate charitable giving literature were looked at, conducted on common terms and variations of words associated with related words. Second, we refine the research scope of the literature and divide all the literature into seven research topics according to different types of research, which are Corporate Charitable Giving and Performance, Corporate Charitable Giving and Innovation, Corporate Charitable Giving and Corporate Social Responsibility, Factors influencing corporate charitable giving, The Impact of Corporate Charitable Giving on Business, A few literature reviews related to corporate charitable giving and others. The proportion of different research contents is shown in Figure1.

Figure1. Classification of research related to corporate charitable giving

Data analysis

As for corporate donation, previous studies have paid more attention to the effectiveness of corporate donation and the different effects of corporate donation on consumer behavior. Early researchers (Fry et al., 1982) [1] found that corporate philanthropy provides direct benefits to participants and benefits to individuals, organizations and communities in the external business environment.

Although there is relatively little direct evidence in corporate charity literature, a large number of marketing studies related to causes show that the contribution of enterprises has a significant positive impact on consumers' perception. Specifically, when a company donates more money to charity, consumers will have a more positive evaluation of the company. For example, Pracejus et al. (2003) [2]found that the donation amount has a positive impact on product selection when the price is constant. Similarly, the research of Haruvy et al. (2009) [3]shows that the donation amount of charity auction (expressed as a percentage of auction income) has a positive impact on auction income.

In previous studies, a large number of scholars have studied donation strategies. For example, Hildebrand (2017) [4]proved in the research that the evaluation advantages given by in-kind contributions actually depend on the characteristics of CSR problems caused by contributions, such as their perceived controllability, by comparing consumers' preferences for two kinds of enterprises-monetary donations and in-kind donations, and through scenario experiments and data analysis. Jin (2018) [5] put forward two new donation strategies, focusing on the amount or frequency of corporate donations, and verified that consumers have different cognitive and behavioral responses to enterprises adopting different donation strategies. Krishna and Rajan (2009) [6] found that even in a competitive environment, charity-related promotions can improve profitability, and in fact, even for those products directly involved in promotions, customers' willingness to pay for company products can be improved.

The next section presents a comprehensive analysis and review about Industry practice and theoretical development of corporate charitable giving, and then reveals the multi-level mechanisms and paths of charitable giving influencing innovation from different perspectives.

Methods 

Suppose authors used the Systematic literature review protocol. Please write exclusion and inclusion criteria in the methodology section.  Please clarify and justify. In the review article, we have a methodology section where we discuss how you conducted the systematic review protocol.  Include this section after the introduction.  

  Response : The response to this question is as described above.

Data Analysis 

Please improve the quality of this section. I can’t see the tables and charts we usually do during the systematic review. You need to justify your findings and how you reviewed the papers. 

  Response : Thank you for your advice, I have added a part about how I collect and review this reseach.  

Discussion and findings 

This title is missing; there is discussion but unguided. Please revise and include a discussion section on how you concluded and came up with a proposed framework. 

  Response : Thank you for your advice, I have added the discussion in the last part, and I raised the framework from related literatures’ opinion, and combined with present corporate charitable giving research.

Conclusion 

 This section should be at the end of your paper. You can discuss research limitations that are missing and future call after discussion.  The conclusion section should only conclude your findings

 Response : Ok, I have reduced this part to mainly about my findings and discussion.

Reference:

  • Fry L W, Keim G D, Meiners R E. Corporate contributions: Altruistic or for-profit[J]? The Academy of Management Journal, 1982, 25(1):94–106.
  • Pracejus J, Deng Q, Olsen G, Messinger P. Fit in cause-related marketing: An integrative retrospective[J]. Journal of Global Scholars of Marketing Science, 2020, 30(2):105–114.
  • Haruvy E, Leszczyc P. Bidder motives in cause-related auctions[J]. International Journal of Research in Marketing, 2009, 26(4):324-331.
  • Hildebrand D, Demotta Y, Sen S, Valenzuela A. Consumer Responses to Corporate Social Responsibility (CSR) Contribution Type[J]. Journal of Consumer Research, 2017, 44(4):738-758.
  • Jin L, He Y. How the frequency and amount of corporate donations affect consumer perception and behavioral responses[J]. Journal of the Academy of Marketing Science, 2018, 46(6):1072-1088.
  • Krishna A, Rajan U. Cause Marketing: Spillover Effects of Cause-Related Products in a Product Portfolio[J]. Management Science, 55(9):1469-1485.

Reviewer 3 Report

This paper provides a literature review on the topics of corporate philanthropy and corporate innovation. However, there are several limitations in performing the review:

What is the methodology for doing this review?

Did the authors use a systematic review? How the following steps are conducted:

·        Search for relevant studies

·        Evaluate references

·        Identify themes and gaps.

·        Outline the structure.

·        Write your literature review.

How 36 papers are selected among 196 papers?

What is the contribution of this study?

How the proposed theoretical perspective is validated?

It seems that this version is a preliminary version of research (under development) that can be published at a conference. So, it is not ready for journal publication.

Author Response

Response to Reviewer 3 Comments

This paper provides a literature review on the topics of corporate philanthropy and corporate innovation. However, there are several limitations in performing the review:

What is the methodology for doing this review?
 Response : Literature analysis methodology.

Did the authors use a systematic review? How the following steps are conducted:

· Search for relevant studies

· Evaluate references

· Identify themes and gaps.

· Outline the structure.

· Write your literature review.
 Response : 
We collected relevant studies and divided into seven different themes, then find the research gap and confine our research questions.

How 36 papers are selected among 196 papers?
 Response : 
To be the very first step, we used data collection to delineate the boundaries of our literature review. Data collection of papers and documents was published between 1950 and 2021. First, we searched major business database, such as ScienceDirect, ProQuest, Springer Link, Emerald and so on. Corporate charitable giving literature were looked at, conducted on common terms and variations of words associated with related words. Second, we refine the research scope of the literature and divide all the literature into seven research topics according to different types of research, which are Corporate Charitable Giving and Performance, Corporate Charitable Giving and Innovation, Corporate Charitable Giving and Corporate Social Responsibility, Factors influencing corporate charitable giving, The Impact of Corporate Charitable Giving on Business, A few literature reviews related to corporate charitable giving and others. The proportion of different research contents is shown in Figure1.

Figure1. Classification of research related to corporate charitable giving
Data analysis
As for corporate donation, previous studies have paid more attention to the effectiveness of corporate donation and the different effects of corporate donation on consumer behavior. Early researchers (Fry et al., 1982) [1] found that corporate philanthropy provides direct benefits to participants and benefits to individuals, organizations and communities in the external business environment.
Although there is relatively little direct evidence in corporate charity literature, a large number of marketing studies related to causes show that the contribution of enterprises has a significant positive impact on consumers' perception. Specifically, when a company donates more money to charity, consumers will have a more positive evaluation of the company. For example, Pracejus et al. (2003) [2]found that the donation amount has a positive impact on product selection when the price is constant. Similarly, the research of Haruvy et al. (2009) [3]shows that the donation amount of charity auction (expressed as a percentage of auction income) has a positive impact on auction income.
In previous studies, a large number of scholars have studied donation strategies. For example, Hildebrand (2017) [4]proved in the research that the evaluation advantages given by in-kind contributions actually depend on the characteristics of CSR problems caused by contributions, such as their perceived controllability, by comparing consumers' preferences for two kinds of enterprises-monetary donations and in-kind donations, and through scenario experiments and data analysis. Jin (2018) [5] put forward two new donation strategies, focusing on the amount or frequency of corporate donations, and verified that consumers have different cognitive and behavioral responses to enterprises adopting different donation strategies. Krishna and Rajan (2009) [6] found that even in a competitive environment, charity-related promotions can improve profitability, and in fact, even for those products directly involved in promotions, customers' willingness to pay for company products can be improved.
The next section presents a comprehensive analysis and review about Industry practice and theoretical development of corporate charitable giving, and then reveals the multi-level mechanisms and paths of charitable giving influencing innovation from different perspectives.

What is the contribution of this study?
 Response : The paper connected corporate charitable giving with enterprise innovation, developed a multi- level and multidisciplinary framework which reveals the fundamental mechanism and the influence path of charitable giving on innovation, which fill the research gap of the theoretical mechanism between charity giving and enterprise innovation. By providing a systematic perspective and theoretical foundation, this research helps to improve the depth and breadth of great significance for future research in the field. In addition, this paper provide businesses with an invaluable suggestion about how to improve their enterprise innovation performance with corporate charitable giving , and through which specific paths.

How the proposed theoretical perspective is validated?
Response : By quoting the verified conclusions in the references, and sorting them out and synthesizing them.

It seems that this version is a preliminary version of research (under development) that can be published at a conference. So, it is not ready for journal publication.

Reference:
[1]    Fry L W, Keim G D, Meiners R E. Corporate contributions: Altruistic or for-profit[J]? The Academy of Management Journal, 1982, 25(1):94–106.
[2]    Pracejus J, Deng Q, Olsen G, Messinger P. Fit in cause-related marketing: An integrative retrospective[J]. Journal of Global Scholars of Marketing Science, 2020, 30(2):105–114.
[3]    Haruvy E, Leszczyc P. Bidder motives in cause-related auctions[J]. International Journal of Research in Marketing, 2009, 26(4):324-331. 
[4]    Hildebrand D, Demotta Y, Sen S, Valenzuela A. Consumer Responses to Corporate Social Responsibility (CSR) Contribution Type[J]. Journal of Consumer Research, 2017, 44(4):738-758.
[5]    Jin L, He Y. How the frequency and amount of corporate donations affect consumer perception and behavioral responses[J]. Journal of the Academy of Marketing Science, 2018, 46(6):1072-1088. 
[6]    Krishna A, Rajan U. Cause Marketing: Spillover Effects of Cause-Related Products in a Product Portfolio[J]. Management Science, 55(9):1469-1485.

Round 2

Reviewer 1 Report

I have one comment left in Section 2.

This section talked about method. However, in Section 2.1 (Data analysis), the content seems to be a review of previous studies. It was not a method for the data analysis.

Please re-check in Section 2.1.

Reviewer 2 Report

Response to Reviewer 2 Comments

The authors have partially improved the paper's quality but still need to improve important sections, including literature, methods, and discussion. The following corrections must be made to improve the quality of this research before publication. (Please read a good review article to see how to summarize results through charts and tables. 

Systematic Literature Review 

Have you used the Systematic literature review protocol? If yes, please summarize the significant findings in table form and show the distribution of journals and details of studies used for the literature review (n~196). However, the complete references show only 36 studies. Please verify and validate your argument. (Review 1)

I still can't see any table that summarizes your reviewed studies. Which journals contain how many articles? Must include those charts etc. it still looks like a traditional review on the topic under consideration. You must not mention 196 articles reviewed while your full references are only 36. there is no clarity on this, please clarify.

Methods 

Suppose authors used the Systematic literature review protocol. Please write exclusion and inclusion criteria in the methodology section. Please clarify and justify.  how many papers exactly have you reviewed? 196 or 36? your references are not justifying this claim. you mentioned a lot of data bases but how you choose 196 can't see details process and elaboration to it.

Discussion and findings 

This title is missing; there is discussion but unguided. Please revise and include a discussion section on how you concluded and came up with a proposed framework. ( Review 1)

I want you to include a separate section titled Discussion and findings. YOU had a discussion but with no title is given. Please address the review 1 comment. 

Reviewer 3 Report

Despite adding some explanation about the method and target studies, the paper is not ready for publication. A concrete methodology for a systematic review is needed to highlight the added value of the review.

Content analysis is an essential part of a review paper. Hene, what is the purpose of section 2.1?

What type of data analysis is conducted?

What keywords are searched? What are the steps for conducting this review?

How the quality of the target studies are evaluated?